# A Leopard Cannot Change Its Spots: Unexpected Products from the Vilsmeier Reaction on 5,10,15-Tritolylcorrole

**DOI:** 10.3390/molecules25163583

**Published:** 2020-08-06

**Authors:** Fabrizio Caroleo, Greta Petrella, Lorena Di Zazzo, Sara Nardis, Beatrice Berionni Berna, Daniel O. Cicero, Roberto Paolesse

**Affiliations:** Dipartimento di Scienze e Tecnologie Chimiche, Università di Roma Tor Vergata, via della Ricerca Scientifica, 1, 00133 Rome, Italy; fabrizio.caroleo@uniroma2.it (F.C.); petrella@scienze.uniroma2.it (G.P.); dizazzo_94@libero.it (L.D.Z.); nardis@scienze.uniroma2.it (S.N.); beatrice.bberna@gmail.com (B.B.B.); cicero@scienze.uniroma2.it (D.O.C.)

**Keywords:** corrole, isocorrole, Vilsmeier reaction, porphyrinoids, NMR

## Abstract

The reaction of 5,10,15-tritolylcorrole with 3-dimethylaminoacrolein (3-DMA) and POCl_3_ gives a further example of the rebel reactivity of this contracted macrocycle. While no evidence was obtained for the formation of the expected β-acrolein corrole, the inner core substituted N21,N22-3-formylpropylcorrole and the 10-acrolein isocorrole were the reaction products. By increasing the temperature or the amount of the Vilsmeier reagent, the 10-isocorrole became the unique reaction product. The formation of the isocorrole by electrophilic attack of the Vilsmeier reagent to the 10-position of the corrole is unprecedented in the porphyrinoids field and it could pave the way for a novel route to the preparation of stable isocorroles.

## 1. Introduction

Corrole is one of the first synthetic porphyrin analogues reported in the literature, being one of the results of the Johnson’s group approach to Vitamin B12 synthesis [1]. The investigations on this macrocycle have been mitigated for a long time by the quite laborious synthetic pathway of β-alkylcorroles [2], and from its intriguing reactivity, which from the beginning was the spring of a series of erroneous interpretations of corrole derivatives [3]. Even the first report on corrole structure was not correct, but later amended by Johnson [4]. The development of simple synthetic routes for the preparation of 5,10,15-triarylcorroles from commercially available starting reagents [5,6,7] has allowed a more detailed investigation of the chemistry of such a macrocycle and, consequently, a deeper understanding of its behavior [8]. The contracted molecular skeleton and the trianionic nature as ligand are two important characteristics that make corrole different with respect to the porphyrin counterparts, as well as its non-innocent character as ligand [9] or its facile oxidation [10].

These differences also influence the corrole reactivity and its functionalization pathways: corroles for example are more reactive towards electrophilic reagents than porphyrins, allowing both the reaction on the corrole free base and a different regioselectivity. This feature can be an advantage because the macrocycle is not inactivated by core protonation [11], thus avoiding the need of metal complex intermediate, although the reaction is always oriented on the macrocycle β-positions [12].

This regioselectivity can become a nuisance for a straightforward preparation of *meso*-sulfonato- or *meso*-nitrophenylcorroles, in contrast to what can be easily achieved with porphyrins [13].

One example of this peculiar reactivity of free base corroles is given by the Vilsmeier reaction: in this case, the product from the reaction of 2,3,7,13,17,18-hexamethyl,8,12-diethylcorrole **1** is a dimethylamminomethene **2a** instead of the expected 5-formyl or 10-formyl derivatives **2b**,**c** (Scheme 1) [14].

In the case of 5,10,15-triphenylcorrole **3**, this macrocycle reveals once again an unusual reactivity: the Vilsmeier reagent attacks also the macrocycle inner core, giving a mixture of the 3-formyl derivative **4a** and the inner core ethane bridged derivative **4b**, reported in Scheme 2 [11].

Recently, the synthesis of acrolein-substituted corroles via Vilsmeier reaction has been reported with very good yields. Thanks to its synthetic versatility, this substituent has been successfully used, in a cobalt corrole derivative, as hybrid material for sensing applications [15], and to finely tune the photophysical properties of gallium derivatives, inducing significant red shifted bright emissions in the NIR region, which make these complexes interesting for optical imaging purposes [16]. However, the metal ion in the inner core strongly influences the reaction success, as demonstrated by the low yields obtained in the case of Cu derivatives [16]. The direct functionalization of the corrole free base could allow for overcoming this drawback, considering the reactivity of corrole under the Vilsmeier reaction conditions. To test the effectiveness of the acrolein substitution on free base corroles, 5,10,15-tritolylcorrole **5** was prepared and used as starting material for the Vilsmeier reaction. Once again, corrole revealed its rebel nature showing a surprising reactivity, leading to unexpected products.

## 2. Results and Discussion

In our previous studies on the Vilsmeier reaction on free base corroles, we observed a significant decomposition of the starting material [15]; it has thus been envisaged to first tackle the decomposition challenge by a modified synthetic protocol, forming the Vilsmeier reagent in CH_2_Cl_2_ and then adding it dropwise to a CH_2_Cl_2_ solution of **5** at 0 °C, in a corrole/reagent molar ratio of 1:90. The solution was left to warm up to room temperature. This procedure reduced the corrole decomposition, but we observed only traces of product with unreacted **5** after 7 h of reaction. To maximize the substrate conversion, we heated the solution at 35 °C for 7 h, maintaining the same corrole/reagent ratio. In this case, **5** fully reacted and formed two main products in satisfying yields (Scheme 3).

Chromatographic separation on silica gel (CH_2_Cl_2_/hexane 1:1) allowed the isolation of the reaction products. UV-vis spectrum of the first fraction showed a corrole free base pattern (Figure 1), with a mass spectrum indicating the presence of an acrolein group on the macrocyclic skeleton (Appendix A).

A detailed NMR investigation was deemed necessary for a thorough characterization of the product: the presence in the ^1^H-NMR spectrum of the resonances corresponding to eight β-pyrrolic protons, together with highfield-shifted signals, suggested an inner core substitution.

Figure 2 shows the ^1^H-NMR spectrum of **6**. The region containing the aromatic signals of the tolyl groups shows three signals with one-proton integration, indicating the loss of symmetry in one tolyl ring; moreover, the presence of signals between −1.5 and −2.7 ppm is compatible with an inner core substitution.

The ^1^H-^13^C HSQC spectrum (Figure 3a) provided further evidence of the presence of eight different β-pyrrolic signals, indicating that no substitution occurred in those positions. The regions containing tolyl signals confirm that of one ring presenting two different signals for protons in position 2 and 3, thus indicating an asymmetry in the structure. ^1^H-^1^H DQFCOSY and NOESY experiments (Figure 3b,c) allowed the assignment of all β-proton signals, together with the three tolyl groups. This analysis identified the tolyl ring attached to position 10 as the one presenting the two non-equivalent faces. Cross peak intensities obtained from NOESY experiments at variable mixing times were fitted to extract the apparent kinetic rotational rate constant (*k*^10^) for this tolyl ring. A faster rotation of protons in the 5 and 15 *meso-*rings results in a single signal, in line with what already observed for ortho-substituted phenyl corroles, which show in all cases *k*^10^ < *k*^5,15^ [17].

Data obtained so far indicate the presence of a substituent located in the inner part of the macrocycle, making the upper and lower face of the ring plane not equivalent (Figure 4a). To further characterize the nature of this substituent, we analyzed the COSY cross peaks at 6.26, −1.57, −1.70 and −2.68 ppm (Figure 4a), together with a multiplicity edited version of ^1^H-^13^C HSQC (Figure 4b), which gives information about the number of protons attached to each carbon.

Combining the information obtained by ^1^H-^1^H DQFCOSY and ^1^H-^13^C HSQC, it was possible to elucidate the structure of the inner core substituent (Figure 4). Carbon chemical shift allows for assigning the signal at 6.26 ppm to an aldehyde (C at 193.1 ppm). Coupling constants between the aldehyde and the CH_2_ at −1.57 and −1.70 ppm are low as expected (1.5 and 3.0 Hz). The CH_2_ group shows a cross peak with opposite sign to the other two CH in the multiplicity-edited HSQC experiment and is coupled to the other CH group (*J* of 5.7 and 7.0 Hz). This last CH group presents a carbon chemical shift of 52.7 ppm, compatible with its position between two inner core nitrogens of the corrole.

To determine the connection between the inner core substituent and the corrole, we have performed a ^1^H-^13^C HMBC experiment (Figure 5).

Cross peaks in the aromatic region of the ^1^H-^13^C HMBC (Figure 5) allowed the assignment of most of the carbons of **6**. The proton at −2.68 ppm showed two correlations with C4 and C6/9. Based on this information, the proposed structure of **6** contains the CH group as a bridge between the two pyrrolic rings flanking the *meso* C-5. Table 1 shows the assignment of ^1^H and ^13^C nuclei of **6**.

The formation of **6** is due to the attack of the electrophile to the macrocycle inner core: this result is not unprecedented in the case of corroles, since we already observed a similar behavior in the case of Vilsmeier formylation [11], and a similar inner core functionalization upon reaction of phosgene with pentafluorophenylcorrole was previously reported [18].

The characterization of the second fraction gave an even more surprising result. The mass spectrum indicated again the introduction of an acrolein substituent in the macrocycle (Appendix A), but the UV-vis showed a different pattern, with two broad Q bands bathochromically shifted (Figure 6).

The ^1^H-NMR spectrum (Figure 7) showed a resonance peak at very high chemical shifts, around 14 ppm, indicating a loss of aromaticity in the system, typical in isocorrole species [19]. The presence of β-pyrrolic proton signals between 6.0 and 6.9 ppm provided a further hint on the lack of conjugation in the system. The presence of four signals in the substituting moiety, each integrating for two protons, indicates that the substituent is not bound to a pyrrolic ring and that the compound presents a symmetric structure.

The nature of the substituent became clear using a combination of ^1^H-^1^H COSY and ^1^H-^13^C HSQC experiments (Figure 8). In the latter, no negative peaks were observed, indicating the lack of CH_2_ groups in the molecule. The presence of a signal at 9.63 ppm attached to a carbon at 193.2 ppm clearly indicates the presence of an aldehyde. The aldehydic proton is coupled to a proton belonging to an HC=CH system, based on their ^1^H and ^13^C chemical shifts. In this way, the substituent was identified as an acrolein moiety.

The absence of aromaticity and the symmetry of the structure is only compatible with a substitution on C-10. This was confirmed by the analysis of cross peaks in the ^1^H-^13^C HMBC and ^1^H-^1^H ROESY experiments (Figure 9).

C10 shows a chemical shift of 55.0 ppm, which is typical of a sp^3^ carbons. The HMBC experiment shows the coupling between H2a and H1a of the acrolein substituent with C10, and that of H1a with C9/11 of the pyrrole. The ROESY spectrum proves the proximity of H1a of the acrolein to both H8/12 and H2′ of the tolyl substituent attached to C10. Analysis of all the 2D experiments provided the thorough assignment of compound **7** (Table 2).

The formation of the isocorrole is due to the electrophilic attack of the Vilsmeier reagent to the 10-position, with the plausible mechanism reported in Scheme 4.

To the best of our knowledge, this is the first example of this kind of reactivity for *meso*-arylcorroles, opening a different direct synthetic pathway for the preparation of *meso*-alkyl isocorroles. We have previously reported the preparation of *meso*-alkyl isocorroles by reaction of Grignard reagent with oxidized corroles [20].

It is interesting to note that this reaction signs a further difference of corroles to respect the parent porphyrins: in the case of corroles, isocorroles are obtained by electrophilic attack to the *meso* positions, while, in the case of porphyrins, phlorins can be obtained by reaction with nucleophiles, like BuLi [21]. This difference proves once more the electron richness of corrole, one of the characteristic features of this macrocycle.

It is also worth mentioning that isocorrole **7** (28%) became the only reaction product when the reaction was carried out at higher temperature in dichloroethane (60 °C) or increasing the amount of the Vilsmeier reagent (1:135 molar ratio respect to **5**). In both cases, the unique formation of isocorrole can be reasonably ascribed to the increased formation of the protonated form of the starting corrole, evidenced by the UV-vis spectrum of the reaction mixture after addition of the Vilsmeier reagent, which precludes its attack to the macrocycle inner core.

Notably, the reaction is highly regioselective: only 10-isocorrole was obtained, without any formation of the 5-isomer, which is usually observed in the other reaction pathways [16]. The origin of this regioselectivity cannot be clearly individuated; however, it should be noted that it is not unprecedented because, also in the case of octaalkylcorrole, it was observed the regioselective attack of the Vilsmeier reagent to the 10-position of corrole, leading to the product reported in Scheme 1 [14].

We have then studied the insertion of metal ions in the obtained isocorrole, with the aim of investigating its stability and consequently its role as precursor for a wide range of metal complexes, which have recently arisen interest as NIR dyes. Among the different metal ions, we choose nickel; the corresponding metal complex is diamagnetic, allowing an easier characterization. Furthermore, Ni isocorrole complexes have been already reported and their properties recently investigated [22].

The reaction was carried out by adding Ni(OAc)_2_ to a solution of **7** in DMF, and the final mixture was stirred at reflux for 1 h. The crude product was purified on preparative TLC (silica gel). Elution with dichloromethane/hexane (2:1) afforded a brown fraction as the major product (34%), along with trace amounts of the starting material **7** (Scheme 5).

The UV-vis showed the characteristic absorption of a metal isocorrole, with an NIR absorption band at 882 nm (Figure 10).

The ^1^H-NMR spectrum of the product did not show the resonance signal at δ = 14 (Appendix A), indicating the restoration of the aromaticity through the nickel complexation; moreover, the distinctive resonance signals of acrolein group allowed the identification of compound **8**.

## 3. Materials and Methods

Reagents and solvents (Aldrich) were of the highest grade available and were used without further purification. Thin-layer chromatography (TLC) was performed on Sigma-Aldrich silica gel plates. Chromatographic purification of the reaction products was accomplished by using silica gel 60 (70–230 mesh, Sigma-Aldrich, St. Louis, MO, USA) as a stationary phase. UV-vis spectra were measured on a Varian Cary 50 Spectrophotometer using CH_2_Cl_2_ as solvent.

NMR experiments were performed in deuterated acetone at 15 °C and recorded with a Bruker Avance spectrometer operating at 700 MHz for ^1^H, equipped with a 5 mm inverse TXI probe and *z*-axis gradients. The ^1^H-^1^H COSY experiment was acquired with a spectral window of 20 × 15 ppm (carrier frequency at 3.5 ppm) using 4096 × 1094 data points and 2 transients. The ^1^H-^1^H ROESY experiment was acquired with a spectral window of 15 × 10 ppm (carrier frequency at 5 ppm) using 4096 × 512 data points, 4 transients and a mixing time of 800 ms. The ^1^H-^1^H NOESY experiment was acquired with a spectral window of 20 × 15 ppm (carrier frequency at 3.5 ppm) using 2048 × 512 data points, 4 transients and a mixing time of 800 ms. ^1^H-^13^C HSQC experiments were acquired with a spectral window of 20 ppm × 33 ppm (carrier frequencies at 3.50 and 121.5 ppm) using 4096 × 256 data points, 8 transients for the aromatic region; 20 ppm × 10 ppm (carrier frequencies at 3.50 and 193.8 ppm) using 4096 × 64 data points, 8 transients for the carbonyl region. Multiplicity-edited ^1^H-^13^C HSQC was acquired with a spectral window of 20 ppm × 36 ppm (carrier frequencies at 3.50 and 36 ppm) using 4096 × 128 data points, 16 transients. ^1^H-^13^C HMBC experiments were acquired with a spectral window of 20 ppm × 190 ppm (carrier frequency at 3.50 and 105.0 ppm) using 4096 × 1024 data points and 16 transients. All experiments were acquired with a relaxation delay of 2 s. All data were processed with TopSpin using a sine squared window function for each dimension of the 2D data.

Apparent kinetic rotational rate constant (*k*^10^) for 10 tolyl ring has been calculated by fitting exchange peaks intensities between signals at 8.318 and 7.798 ppm. Each NOESY experiment was acquired with a spectral window of 20 × 15 ppm (carrier frequency at 3.5 ppm) using 2048 × 512 data points, 4 transients and at 10 different mixing times: 0.05 s, 0.2 s, 0.3 s, 0.4 s, 0.5 s, 0.6 s, 0.7 s, 0.8 s, 1 s, 1.2 s, 1.5 s:

$$I=A\times \sinh\left(\tau_{m}\times k^{10}\right)\times exp^{\left(-\left(k^{10}+T_{1}^{-1}\right)\tau_{m}\right)}$$where *A* is an intensity factor, *τ_m_* in the mixing time, and *T*_1_ the longitudinal relaxation time.

### 3.1. General Procedure for the Vilsmeier Reaction of ***5***

The Vilsmeier reagent was prepared by reacting 3-DMA (1.43 mL, 15.57 mmol) and adding POCl_3_ (1.33 mL, 15.75 mmol) under nitrogen at 0 °C; the reagent was then added dropwise to a solution of **5** [23] (100 mg; 0.17 mmol) in CH_2_Cl_2_ (70 mL). The resulting mixture was stirred at 35 °C under nitrogen. The progress of the reaction was monitored by UV-vis spectroscopy; after seven hours, when no more starting material was detected, a saturated solution of NaHCO_3_ (130 mL) was added and the mixture was stirred overnight at room temperature. The organic phase was separated, washed with brine and then twice with water, then dried on anhydrous sodium sulfate. The crude mixture was purified by column chromatography (silica gel, elution with 1:1 mixture of CH_2_Cl_2_/Hexane), affording two fractions. Compound **6** eluted first as main product (*R*_f_ = 0.54, 46 mg, 43% yield), then **7** as second fraction (*R*_f_ = 0.41, 18 mg, 18% yield).

When the same reaction was carried out using a larger amount of 3-DMA/POCl_3_ Vilsmeier reagent (1:135 molar ratio respect to **5**) or at higher temperature (60 °C in C_2_H_4_Cl_2_), the subsequent chromatographic workup afforded only **7** (30 mg, 28% yield).

**6**: UV-vis (CH_2_Cl_2_), *λ*_max_ [nm] (*ε*): 424 (3.15 × 10^4^), 568 (4.32 × 10^3^), 618 (4.06 × 10^3^). Anal. Calcd for C_43_H_34_N_4_O: C, 82.9; H, 5.5; N, 9.0. Found: C, 83.0; H, 5.6; N, 8.9%. MS (MALDI-TOF): *m*/*z* calcd. for C_43_H_34_N_4_O 622.27, found 622.54. ^1^H-NMR (700 MHz, deuterated acetone): δ = 9.5 (d, 1H, *J* = 4.3 Hz), 9.19(d, 1H, *J* = 4.0 Hz), 8.870 (d, 1H, *J* = 4.04 Hz), 8.750(d, 1H, *J* = 4.59 Hz), 8.739 (d, 1H, *J* = 2.6 Hz), 8.71 (d, 1H, *J* = 4.3 Hz), 8.503 (d, 1H, *J* = 4.9 Hz), 8.320 (d, 1H, *J* = 0.4 Hz), 8.318 (d, 2H, *J* = 0.4 Hz), 8.31 (s, 1H, *J* = 0.4 Hz),8.27 (d, 2H, *J* = 0.4 Hz), 7.79 (s, 1H, *J* = 0.4 Hz), 7.72 (d, 2H, *J* = 0.4 Hz), 7.72 (d, 2H, *J* = 0.4 Hz), 7.70 (s, 1H, *J* = 0.4 Hz),7.5 (s, 1H, *J* = 6.9 Hz), 6.26 (t, 1H, *J* = 0.4 Hz), 2.69 (s, 3H, *J* = 18.3 Hz), 2.67 (s, 3H, *J* = 11.1 Hz), 2.6 (s, 3H, *J* = 10 Hz), −1.56 (d, 2H, 16.6, 6.9, 2.2 Hz), −1.7 (d, 2H, *J* = 0.4 Hz), −2.681 (t, 1H, *J* = 6.2 Hz). ^13^C-NMR (100 MHz, deuterated acetone): δ = 193.0, 143.1, 141.1, 141.0, 138.7, 137.1, 136.7, 136.2, 134.8, 134.5, 134.1, 133.8, 133.7, 133.5, 132.5, 132.4, 131.3, 129.1, 128.6, 128.2, 127.2, 126.4, 125.9, 125.1, 124.8, 121.2, 118.0, 117.3, 113.0, 112.6, 110.8, 52.8, 41.6, 20.1, 20.0.

**7**: UV-vis (CH_2_Cl_2_), *λ*_max_ [nm] (*ε*): 436 (3.33 × 10^4^), 655 (2.42 × 10^3^), 706 (2.74 × 10^3^). Anal. Calcd for C_43_H_34_N_4_O: C, 82.9; H, 5.5; N, 9.0. Found: C, 82.7; H, 5.6; N, 9.2%. MS (MALDI-TOF): MS (MALDI-TOF): *m*/*z* calcd. for C_43_H_34_N_4_O 622.27, found 622.07. ^1^H-NMR (700 MHz, deuterated acetone): δ = 14.54 (s, 2H, *J* = 0.4 Hz), 9.67 (d, 1H, *J* = 7.7 Hz), 7.58 (d, 1H, *J* = 0.4 Hz), 7.55 (d, 4H, *J* = 21.8, 11.4 Hz), 7.33 (d, 4H, *J* = 7.7 Hz), 7.25 (d, 2H, *J* = 8.1 Hz), 7.20 (d, 2H, *J* = 8 Hz), 6.84 (dd, 4H, *J* = 4.1 Hz), 6.73 (d, 2H, *J* = 4.2 Hz), 6.01 (d, 2H, *J* = 4.2 Hz), 5.9 (dd,1H, *J* = 15.7, 7.7 Hz), 2.53 (s, 6H, *J* = 0.4 Hz), 2.43 (2, 3H, *J* = 0.4 Hz). ^13^C-NMR (100 MHz, deuterated acetone): δ = ^13^C-NMR (100 MHz, acetone-d_6_) δ = 193.3, 158.9, 145.5, 140.7, 140.3, 140.2, 139.0, 137.7, 136.0, 133.4, 129.5, 128.1, 115.6, 55.0, 20.9, 20.4.

### 3.2. Synthesis of ***8***

**7** (16 mg, 0.025 mmol) was dissolved in DMF (7.5 mL), Ni(OAc)_2_ 4H_2_O (96 mg, 0.36 mmol) was added and the mixture was stirred at reflux for 1 h. The progress of the reaction was monitored by UV/vis and TLC analysis (silica, eluent mixture CH_2_Cl_2_/Hexane 2/1 *v*/*v*). After one hour, there was no evidence of the starting material and the solvent was removed under vacuum. The crude product was dissolved in CH_2_Cl_2_ and purified on preparative TLC (silica, eluent mixture CH_2_Cl_2_/Hexane, 2:1, *R*_f_ = 0.6). A brown fraction corresponding to **8** was obtained as the major product (11 mg, 65% yield), followed by traces of the starting material.

UV-vis (CH_2_Cl_2_): *λ*_max_ [nm] (*ε*) 430 (1.68 × 10^4^), 531 (7.82 × 10^3^), 562 (7.85 × 10^3^), 880 (7.73 × 10^3^). ^1^H**-**NMR (400 MHz, CDCl_3_) δ = 9.65 (d, 1H, *J* = 7.6 Hz), 7.65 (d, 1H, *J* = 15.6 Hz), 7.45(d, 4H, *J* = 10.3, 7.7 Hz),7.29(d, 4, *J* = 0.4 Hz), 7.21(d, 2H, *J* = 0.4 Hz), 7.19 (d, 2H, *J* = 0.4 Hz), 6.71 (d, 2H, *J* = 4.5 Hz), 6,6 (d, 4H, *J* = 4.3 Hz), 6.15 (dd, 1H, *J* = 15.6, 7.6 Hz), 6.09 (d, 2H, *J* = 4.5 Hz), 2.5 (s, 6H, *J* = 0.4 Hz), 2.4 (s, 3H, *J* = 0.4 Hz). Anal. Calcd for C_43_H_32_N_4_ONi: C, 76.0; H, 4.7; N, 8.2. Found: C, 76.2; H, 4.6; N, 8.1%. MS (MALDI-TOF): *m*/*z* calcd. For C_43_H_32_N_4_ONi 678.19 found 678.21 (Appendix A).

## 4. Conclusions

The Vilsmeier reaction of dimethylamino acrolein/POCl_3_ with **5** afforded two different products. The first one derives from the attack of the Vilsmeier reagent to the macrocycle inner core, leading to the bridged N21,N22-3-formylpropyl derivative **6**. This reactivity pattern has been previously reported for both phosgene and analogous Vilsmeier formylation reactions. The second product sheds light on a novel reactivity pattern, which, to the best of our knowledge, is unprecedented for this macrocycle. In this case, the Vilsmeier reagent regioselectively attacks the 10-*meso* position, leading to the formation of the isocorrole derivative **7**.

The isocorrole becomes the unique product by increasing the reaction temperature or the amount of the Vilsmeier reagent; this result can be ascribed to the increased formation of the corrole cation by protonation, which blocks the inner core reactivity.

It should be noted that both the β-pyrrolic and *meso*-phenyl group positions are not reactive, in contrast to what was observed in electrophilic reactions on free base tetraphenylporphyrins. This route could open the way for the preparation of isocorroles and the corresponding metal complexes, and this synthetic approach is currently ongoing in our laboratories.

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
