# Peer review of "A Leopard Cannot Change Its Spots: Unexpected Products from the Vilsmeier Reaction on 5,10,15-Tritolylcorrole"

_molecules, 2020, doi:10.3390/molecules25163583_

Round 1

Reviewer 1 Report

In this manuscript the unprecedented formation of an isocorrole from the Vilsmeier reaction of 5,10,15-tritolylcorrole. The manuscript is relatively well written. Only a few details throughout the manuscript need to be addressed:

  • In Scheme 3 the reaction yields should be included.
  • 13C chemical shifts are reported in tables. Are these from normal 13C or indirectly obtained from the 2D experiments?
  • In Scheme 4 at the 3rd intermediate, the curly arrows are wrong and need to be corrected. The arrow should start from the oxygen and ending at the carbon of the iminium cation.
  • In paragraph with lines 199-202 the yield of the reaction and compound number should be included.
  • In the experimental section in lines 263-265 the compound numbers (1 and 3) are wrong and need to be correct.
  • All the spectroscopic characterisation data should be reported in the experimental section e. 1H/13C chemical shifts etc, despite reported in the main manuscript.
  • In the Supporting Information, all the acquired NMR spectra should be included in full size despite some of them are included in the main manuscript.
  • At line 215 remove the repeated “the” at the beginning of the sentence.

Author Response

1) In Scheme 3 the reaction yields should be included.

The scheme has modified as indicated by the reviewer.

2) 13C chemical shifts are reported in tables. Are these from normal 13C or indirectly obtained from the 2D experiments?

The spectra have been obtained from normal 13C and they have now been reported in the Supplementary Information and listed in the Experimental section.

3) In Scheme 4 at the 3rd intermediate, the curly arrows are wrong and need to be corrected. The arrow should start from the oxygen and ending at the carbon of the iminium cation.

The scheme has modified as indicated by the reviewer.

4)  In paragraph with lines 199-202 the yield of the reaction and compound number should be included

The data have been added.

5) In the experimental section in lines 263-265 the compound numbers (1 and 3) are wrong and need to be correct.

The error has been corrected, now the correct numbers have been used.

6) All the spectroscopic characterisation data should be reported in the experimental section e.1H/13C chemical shifts etc, despite reported in the main manuscript

The NMR data have been added in the Experimental section

7) At line 215 remove the repeated “the” at the beginning of the sentence.

The error has been corrected

Reviewer 2 Report

Specific Comments:

  1. Author should provide J coupling constant values in the spectral data.
  2. Please check typographical errors

Author Response

1) Author should provide J coupling constant values in the spectral data.

J coupling constants have been listed in the Experimental section.

2) Please check typographical errors

The text has been carefully revised.

Reviewer 3 Report

In the manuscript under review, the authors report unprecedented and unexpected reactivity of a corrole derivative resulting in formation of an inner core substituted corrole and an isocorrole. The structure of the isolated products is established based on a thorough NMR study and the presented analysis is quite sound. The manuscript can be of interest to the specialists working in the field of porphyrinoids. After some text and style revision, the manuscript can be recommended for publication.

My remarks:

1) P. 13, line 265 -- seems to be a mistake: how can the Vilsmeier reaction of corrole 5 give corrole 3?

2) The statement in the Conclusions "The isocorrole becomes the unique product by increasing the reaction temperature or the amount of the Vilsmeier reagent; this result can be ascribed to the increased formation of the corrole cation by protonation, which blocks the inner core reactivity" is not disclosed in the maintext of the manuscript. 

3) The possible applications of the compounds under study should be specified more explicitly. The descirptions like "to fine tune photophysical properties of..." or "interesting optical properties" are not so informative.

4) Why did the authors choose nickel among all possible metals? It would be interesting to see the explanation of this choice in the manuscript.

5) There are some typos in the text, like "acroelin", p. 12, line 222.

Author Response

1) P. 13, line 265 -- seems to be a mistake: how can the Vilsmeier reaction of corrole 5 give corrole 3?

The error in compound numbering has been corrected.

2) The statement in the Conclusions "The isocorrole becomes the unique product by increasing the reaction temperature or the amount of the Vilsmeier reagent; this result can be ascribed to the increased formation of the corrole cation by protonation, which blocks the inner core reactivity" is not disclosed in the maintext of the manuscript.

A sentence was present in the main text of the original manuscript (lines 201-202) and it has been now more detailed in the revised text (lines 203-206).  

3) The possible applications of the compounds under study should be specified more explicitly. The descirptions like "to fine tune photophysical properties of..." or "interesting optical properties" are not so informative.

More information has been added as suggested by the reviewer (lines 57-58 and line 215).

4)  Why did the authors choose nickel among all possible metals? It would be interesting to see the explanation of this choice in the manuscript

A sentence containing the explanation of Ni choice has been added to the text (lines 215-217).

5) There are some typos in the text, like "acroelin", p. 12, line 222.

The text has been carefully checked to remove the errors.

Reviewer 4 Report

The present manuscript by Paolesse and co-workers describes the reaction of 5,10,15-tritolylcorrole with 3-dimethylaminoacrolein (3-DMA) and POCl3. The authors claim the regioselective formation of isocorrole; however, the reaction conditions were not optimized. Additionally, a detailed substrate scope is not given. A thorough NMR study was performed to characterize the synthesized compounds. The manuscript is very well-written and interesting, but some points of the points given above and below would improve the manuscript. In conclusion, the paper is recommended for publication after revision.

  1. Page 3 Line 85 ‘By 1H NMR analysis ….... core substitution’ please rephrases the sentence.
  2. Please label the chemical shift values in Figure 2 and Figure 7.
  3. Corrole 6 and 7: 1H NMR data is not included in the experimental section.
  4. Please provide Rf and melting points for synthesized corroles 6, 7, and 8.
  5. The authors should include the DEPT-135 for corroles 6 and 7.
  6. Figure S4: 1H NMR peaks are not integrated. Please integrate the spectra and label the chemical shift values.
  7. Please provide 13C NMR for all synthesized compounds.
  8. Authors claimed that ‘isocorrole became the only reaction product when the reaction was carried out at a higher temperature in dichloroethane or increasing the amount of the Vilsmeier reagent’ however, neither temp nor equiv. are specified in the current manuscript. Please add a suitable explanation/detail.
  9. The formation of isocorrole is highly regioselective. Why?
  10. Please recollect the full range NIR absorption spectra for compound 8.

Author Response

1) Page 3 Line 85 ‘By 1H NMR analysis ….... core substitution’ please rephrases the sentence.

The sentence has been modified as requested (lines 87-88). 

2) Please label the chemical shift values in Figure 2 and Figure 7.

For the sake of clarity, we have labeled the Figures reported in the Supplementary Information containing the full spectra. In the case of Figure 2 and Figure 7 in our opinion the addition of the peak labels will reduce the figure readability.

3) Corrole 6 and 71H NMR data is not included in the experimental section.

The data have been now added.

4)  The authors should include the DEPT-135 for corroles 6 and 7.

The multiplicity-edited HSQC combines the information from the DEPT-135 and the correlation of an HSQC, so it is not necessary to perform this experiment to differentiate CH2 from CH and CH3. We have presented, in Figure 4b, the fact that the only CH2 of corrole 6 is negative, as it would be in a DEPT-135.

For corrole 7, a molecule without any CH2, we have performed also the multiplicity-edited HSQC, but we did not explicitly state so in the text. Now, the phrase in line 162 reads:

“The nature of the substituent became clear using a combination of 1H-1H COSY and multiplicity edited 1H-13C HSQC experiments (Figure 8). In the latter, no negative peaks were observed, indicating the lack of CH2 groups in the molecule.”

Also, the legend for Figure 8 now includes the description of the multiplicity-edited nature of the HSQC:

“Figure 8. (a) Proposed structure of the substituent of the isocorrole. (b) 1H-1H COSY and (c) multiplicity-edited 1H-13C HSQC experiment regions.”

5) Figure S4: 1H NMR peaks are not integrated. Please integrate the spectra and label the chemical shift values.

The full spectra have now been reported as requested.

6) Please provide 13C NMR for all synthesized compounds.

The NMR data have been added in the Experimental section and the full spectra added in the Supplementary Material.

7) Authors claimed that ‘isocorrole became the only reaction product when the reaction was carried out at a higher temperature in dichloroethane or increasing the amount of the Vilsmeier reagent’ however, neither temp nor equiv. are specified in the current manuscript. Please add a suitable explanation/detail.

The experimental data have now been added and a tentative explanation has now been more detailed (lines 203-206).

8) The formation of isocorrole is highly regioselective. Why?

A comment on the reaction regioselectivity has been added in the text (lines 208-212).

9) Please recollect the full range NIR absorption spectra for compound 8.

The spectrum has now been reported as requested.

Round 2

Reviewer 4 Report

The authors have suitable revised the manuscript, it is in fine form and can be published.